# Perceived Eye-Related Symptoms and Influencing Factors in Hospital Nurses

Ok-Hee Cho [1], Haemin Cho [1] and Hyekyung Kim [2,*]

[1] Department of Nursing, College of Nursing and Health, Kongju National University, Gongju 32588, Republic of Korea; ohcho@kongju.ac.kr (O.-H.C.); whgoals32@naver.com (H.C.)
[2] Department of Nursing, Catholic Kwandong University, Gangneung 25601, Republic of Korea
* Correspondence: hk1224@cku.ac.kr; Tel.: +82-33-649-7615; Fax: +82-33-649-7620

**Abstract:** Nurses are at risk of eye discomfort due to the increasing use of visual display terminals and wearing masks, which may worsen eye-related symptoms. This study was conducted in South Korea to identify the factors influencing eye-related symptoms among hospital nurses on/off duty. The study included 154 nurses who completed a self-reported questionnaire that assessed demographic characteristics, perceived health status, dry-eye symptoms, occupational stress, and eye-related symptoms. The results showed that nurses complained of more eye-related symptoms on duty than off duty, with female sex and dry-eye symptoms being the factors influencing eye-related symptoms on duty. On the other hand, computer use time ($\geq$4 h) and dry-eye symptoms were the factors influencing eye-related symptoms off duty. The study suggests that assessing dry-eye symptoms can facilitate early interventions to relieve eye-related symptoms in hospital nurses, and they should pay attention to eye health during working hours as well as off hours.

**Keywords:** nurses; dry eye syndromes; occupational stress





## 1. Introduction

With the increasing adoption of digitized healthcare, the use of visual display terminals (VDTs) in the field of healthcare has gained prominence. The digitization of healthcare has also caused various changes in the field of nursing; the VDT usage time during on-duty hours is increasing with the use of computers and mobile devices to check prescriptions and create nursing records [1]. However, the use of VDTs is associated with various eye diseases ranging from eye discomfort and tiredness to reduced vision in workers [2]. And prolonged VDT usage is a key factor associated with reduced quality of life since it reduces work productivity and causes inconveniences in daily activities [2,3]. The steps taken to stop COVID-19 from spreading have generally led to a lifestyle marked by prolonged durations of indoor activities, which has increased usage of mobile devices and screen displays, even during leisure time [3]. Dry-eye syndrome, which is characterized by decreased tear production and increased evaporation, is mainly attributable to aging and is a risk factor for eye health [4,5]. However, the increasing time spent using smartphones as well as prolonged mask-wearing for infection prevention has resulted in an increase in the prevalence of dry-eye syndrome [2,6]. Continuous mask-wearing can increase the airflow around the eyes, leading to the promotion of eye dryness [7]. In a pre-COVID-19 pandemic study [8], the reported prevalence of moderate or severe dry-eye syndrome among healthcare workers was 54.0%. A subsequent study conducted after the pandemic [7] reported a prevalence of 62%, supporting the claim of an increase in prevalence. However, nurses at general hospitals, who have low awareness of the importance of eye health and related management, tend to underestimate eye-related symptoms (ES) [9].

Most hospital nurses work in shifts and have irregular sleep hours. They are also exposed to various environmental factors that can pose a threat to eye health. These

factors included high temperature in wards, low indoor humidity for infection control and reduced air flow, and the presence of volatile organic compounds such as anesthetic gases [10]. A previous study [11] found that operating room nurses experience reduced blinking and increased tear evaporation due to long hours of high-level concentration. They also reported experiencing eye irritation or reduced vision from surgical lasers or smoke. In another study [12], inappropriate light intensity at nurses' stations was shown to increase eye tiredness, while Hyon et al. (2019) [10] reported that nurses with high occupational stress experienced a high level of ES. These findings suggest that various work-related factors may play a role in the development of ES among nurses. ES originating from eye tiredness to dry-eye syndrome and vision-related problems could pose a threat to patient safety by reducing nurses' work efficiency [2] and increasing the error rate in their work [13]. Therefore, active management is required for eye health in nurses, since it is a factor that can directly affect the health of the nurses themselves as well as the safety of patients. However, only a few studies to date have reported the perceived ES in nurses. Therefore, this study aimed to determine the severity of perceived ES in nurses during on-duty and off-duty hours and identify the influencing factors. More specifically, we first examined the characteristics related to the severity of perceived ES in nurses and then determined the factors influencing eye-related symptoms on duty (ES on duty) and off duty (ES off duty).

## 2. Materials and Methods

### 2.1. Study Design

This study was a cross-sectional descriptive study conducted on general hospital nurses to identify the perceived ES and their influencing factors in these nurses.

### 2.2. Participants

The participants in this study were nurses with a career ≥3 months at a single general hospital in South Korea; the participants were recruited through convenience sampling. The sample size was determined based on the results of a previous study [7] and calculated using G*power 3.1.9. The minimum required sample size was $n = 153$ with the following conditions: significance level = 0.05, power = 0.95, effect size = 0.15 and number of predictors = 7. Considering the drop-out rate, a total of 160 nurses in total were recruited. After excluding six respondents who provided incomplete responses, the final number of participants was 154.

### 2.3. Instruments

#### 2.3.1. General Characteristics

For evaluating participants' general characteristics, 15 variables were investigated; age, sex, married (yes/no), education level, career length, position, department, shift work (yes/no), working environment: temperature/humidity (appropriate/inappropriate), computer usage time (per day), smartphone usage time (per day), glasses or contact lenses (yes/no), diagnosis of dry-eye syndrome (yes/no), vision-correction surgery (LASIK, LASEK, etc.), and self-care for the eyes.

#### 2.3.2. Perceived Health Status

Perceived health status was measured using a tool developed by Speake et al. [14]. The three items of the tool were rated on a 5-point Likert scale from "Very poor" (1) to "Very good" (5). The score range was 3–15, and higher scores indicated higher levels of perceived health status. Cronbach's $\alpha$ was 0.89 in this study.

#### 2.3.3. Dry-Eye Symptoms

Dry-eye symptoms were measured using the Standard Patient Evaluation of Eye Dryness (SPEED) [15] and the Ocular Surface Disease Index (OSDI) [16]. SPEED consists of 12 items across three subcategories: onset of dry-eye symptoms (at this visit, within the past

72 h, and within the past 3 months), frequency, and severity. The frequency of symptoms was rated as "Never" (0), "Sometimes" (1), "Often" (2), and "Constant" (3). The severity of symptoms was rated as "No problems" (0), "Tolerable" (1), "Uncomfortable—irritating but does not interfere with my day" (2), "Bothersome—irritating and interferes with my day" (3), and "Intolerable" (4). The total score was the sum of the frequency and severity scores, while the data on the onset were not reflected in the score. The score range was 0–28, and higher scores indicated higher frequency and severity of dry-eye symptoms. A score $\geq 6$ was classified as indicating dry-eye) [15]. In this study, Cronbach's $\alpha$ was 0.90. The OSDI consisted of 12 items across three subcategories: ocular symptoms (4 items), vision-related functions (5 items), and environmental triggers (3 items). Each item was rated on a 5-point Likert scale from "All the time" (4) to "None of the time" (0), and the following equation was used: OSDI score = (total score of items with responses * 25/number of items with responses). The score range was 0–100, and higher scores indicated higher levels of ocular surface disease. A score $\geq 13$ was classified as indicating dry eye [16]. In this study, Cronbach's $\alpha$ was 0.90.

### 2.3.4. Occupational Stress

Occupational stress was measured using the Korean Occupational Stress Scale developed by Chang et al. [17]. The 24 items of the tool were rated on a 4-point Likert scale from "Strongly disagree" (1) to "Strongly agree" (4). The score range was 24–96, and higher scores indicated higher occupational stress. Cronbach's $\alpha$ was 0.85 in this study.

### 2.3.5. Perceived Eye-Related Symptoms

Perceived ES was measured using the items of the ES domain of the Video Display Terminals Syndrome, developed by Woo et al. [18]. The tool consisted of 12 items, including "Bloodshot eyes", "Eye strain", "Itchy eyes", "Feeling of eye fatigue", rated on a 5-point Likert scale from "Not at all" (1) to "Very" (5). In this study, the participants were guided to mark each level of perceived ES for "on duty" and "off duty" independently. The score ranged from 24 to 120, and higher scores indicated higher levels of perceived ES. Cronbach's $\alpha$ in this study was 0.94 (0.90 for ES on duty and 0.92 for ES off duty).

### 2.4. Data Collection

The overall study flow and the ethical considerations regarding participants were approved by the Institutional Review Board at K University (KNU_IRB_2021-84). The data collection period was August–September 2021. After obtaining permission from the nursing division and related departments at the respective hospitals, signed consent was obtained from each nurse who voluntarily agreed to participate and complied with the disinfection guidelines of the government. The researcher provided explanations on the study purpose and participants, after which participants who submitted the signed consent form were instructed to complete a self-reported questionnaire. The questionnaires were retrieved immediately on completion; the time taken to complete the questionnaire was approximately 15 min, and the participant received coffee coupons worth 10,000 Korean Won.

### 2.5. Statistical Analysis

The collected data were analyzed using the SPSS Win 27.0 program (IBM Corp., Armonk, NY, USA). For the participants' general characteristics and levels of perceived ES and other variables, the number, percentage, mean and standard deviation were obtained. To analyze the differences in perceived ES in relation to the nurses' characteristics, *t*-test and analysis of variance (ANOVA) were used. Scheffe's test was used as the post-hoc test. A multiple regression analysis was performed to identify the factors influencing the ES on and off duty.

## 3. Results

### 3.1. Participants' General Characteristics

The mean age of the participants was 32.8 years (22–58 years); the number of participants aged <30 years was 80 (52.0%). The study population included 136 females (88.3%), and the number of nurses with a spouse was 53 (34.4%). The number of nurses with a diploma or bachelor's degree was 133 (86.4%), and the number of nurses with >10 years of experience was 48 (31.1%). In the nurse position, there were 136 (88.3%) staff members. The number of nurses working in medical/surgical wards was 46 (32.9%), and the number of those working in shifts was 90 (58.4%). In assessments of the temperature and humidity of the working environment, 102 (66.2%) nurses responded that they were "appropriate." The number of nurses with the daily computer usage time $\geq 4$ h was 116 (75.3%), and the number of nurses with the daily smartphone usage time $\geq 4$ h was 71 (46.1%). Glasses and contact lenses were used by 70 (45.5%) nurses, while 40 (26.0%) nurses had been diagnosed with dry-eye syndrome and 54 (35.1%) nurses had received vision-correction surgery. Regarding self-care for the eyes, 47 (30.5%) nurses used protective eyewear, 34 (22.1%) used a supplementary drug (nutritional supplements, artificial tears, etc.), and 23 (14.9%) underwent regular eye exams (Table 1).

**Table 1.** Differences in perceived eye-related symptoms according to participants' general characteristics (n = 154).

| Characteristics | | Total | Perceived Eye-Related Symptoms | |
|---|---|---|---|---|
| | | n (%) | M ± SD | t/F (p) |
| Age | <30 | 80 (52.0) | 62.24 ± 17.50 | 2.78 (0.065) |
| | 30–39 | 40 (26.0) | 61.85 ± 15.16 | |
| | ≥40 | 34 (22.0) | 69.72 ± 14.01 | |
| Gender | male | 18 (11.7) | 50.72 ± 20.50 | −3.73 (<0.001) |
| | female | 136 (88.3) | 65.48 ± 15.04 | |
| Spouse | yes | 53 (34.4) | 65.86 ± 14.71 | −1.13 (0.262) |
| | no | 101 (65.6) | 62.66 ± 17.18 | |
| Education level | diploma, bachelor | 133 (86.4) | 63.81 ± 16.52 | 0.17 (0.868) |
| | master and above | 21 (13.6) | 63.15 ± 16.19 | |
| Experience (years) | <2 | 22 (14.3) | 58.23 ± 16.82 | 1.05 (0.373) |
| | 2–5 | 42 (27.3) | 63.60 ± 17.75 | |
| | 5–10 | 42 (27.3) | 65.21 ± 15.72 | |
| | >10 | 48 (31.1) | 65.13 ± 15.54 | |
| Position | staff nurse | 136 (88.3) | 63.51 ± 16.30 | −0.46 (0.649) |
| | chief nurse | 18 (11.7) | 65.50 ± 17.90 | |
| Department | Medical and surgical wards (a) | 46 (32.9) | 60.98 ± 15.43 | 2.92 (0.036) |
| | COVID-19 ward (b) | 41 (29.3) | 69.46 ± 17.18 | b > a, c |
| | Intensive care unit, operation room, emergency room (c) | 36 (9.1) | 59.83 ± 16.02 | |
| | outpatient clinic (d) | 31 (22.1) | 64.82 ± 15.73 | |
| Shife work | yes | 90 (58.4) | 64.09 ± 16.95 | 0.33 (0.743) |
| | no | 64 (41.6) | 63.19 ± 15.76 | |
| Working environment temperature/humidity | appropriate | 102 (66.2) | 61.12 ± 16.63 | −2.79 (0.006) |
| | inappropriate | 52 (33.8) | 68.82 ± 14.88 | |
| Computer usage time (h/day) | <4 | 38 (24.7) | 57.34 ± 17.57 | −2.83 (0.005) |
| | ≥4 | 116 (75.3) | 65.87 ± 15.52 | |
| Smartphone usage time (h/day) | <4 | 83 (53.9) | 64.93 ± 15.83 | 0.96 (0.341) |
| | ≥4 | 71 (46.1) | 62.37 ± 17.08 | |
| Glasses, Contact lenses | yes | 70 (45.5) | 65.07 ± 16.50 | 1.02 (0.939) |
| | no | 84 (54.5) | 62.56 ± 16.37 | |
| Diagnosis of dry eye syndrome | yes | 40 (26.0) | 69.08 ± 15.53 | −2.44 (0.016) |
| | no | 114 (74.0) | 61.79 ± 16.38 | |
| Vision correction surgery | yes | 54 (35.1) | 66.23 ± 16.96 | 1.38 (0.169) |
| | no | 100 (64.9) | 62.37 ± 16.05 | |

**Table 1.** *Cont.*

| Characteristics | | Total | Perceived Eye-Related Symptoms | |
|---|---|---|---|---|
| | | n (%) | M ± SD | t/F (p) |
| How to self-care for the eyes * | protective eyewear | 47 (30.5) | | |
| | supplementary drug (nutritional supplements, artificial tears) | 34 (22.1) | | |
| | regular eye exams | 23 (14.9) | | |
| | complementary alternative therapy (diet, massage) | 8 (5.2) | | |
| | none | 71 (46.1) | | |

* multiple response.

### 3.2. Level of Research Variables

The mean perceived ES score was 63.72 (33.37 for ES on duty and 30.30 for ES off duty). The score was significantly higher for ES on duty than for ES off duty (t = 6.48, $p < 0.001$). The average SPEED score was 10.07, and 111 (72.1%) nurses had scores ≥6. The average OSDI score was 25.68, and 36 (23.4%) nurses had a scores ≥13. The mean score for perceived health status was 8.62 and that for occupational stress was 57.30 (Table 2).

**Table 2.** Level of Perceived Health Status, Dry-Eye Symptoms, Occupational Stress, and Perceived Eye-Related Symptoms (n = 154).

| Variables | | n (%) or M ± SD | t (p) |
|---|---|---|---|
| Perceived eye-related symptoms | | 63.72 ± 16.42 | |
| ES-on duty (A) | | 33.37 ± 8.61 | |
| ES-off duty (B) | | 30.30 ± 8.82 | |
| difference (A-B) | | 3.11 ± 5.90 | 6.48 (<0.001) |
| SPEED | | 10.07 ± 5.45 | |
| | <6 | 43 (27.9) | |
| | ≥6 | 111 (72.1) | |
| OSDI | | 25.68 ± 17.53 | |
| | <13 | 118 (76.6) | |
| | ≥13 | 36 (23.4) | |
| Perceived health condition | | 8.62 ± 2.15 | |
| Occupational stress | | 57.30 ± 8.13 | |

ES-on duty = eye-related symptoms during on duty. ES-off duty = eye-related symptoms during off duty. SPEED = Standard Patient Evaluation of Eye Dryness. OSDI = Ocular Surface Disease Index.

### 3.3. Differences in Perceived ES According to General Characteristics

The perceived ES varied according to the participants' sex (t = −3.73, $p < 0.001$), department (F = 2.92, $p = 0.036$), appropriateness of the temperature and humidity of the working environment (t = −2.79, $p = 0.006$), computer usage time per day (t = −2.83, $p < 0.005$), and diagnosis of dry-eye syndrome (t = −2.44, $p = 0.016$) (Table 1).

### 3.4. Factors Influencing the ES on and off Duty

The variables of general characteristics that showed significant differences in relation to perceived ES in the univariate analysis, namely, sex, department, appropriateness of the temperature and humidity of the working environment, computer usage time, and diagnosis of dry-eye syndrome, were applied as independent variables in addition to dry-eye symptoms (SPEED and OSDI scores), occupational stress, and perceived health status in a multiple regression analysis. The factors influencing the ES on duty were female sex (β = 3.11, $p = 0.002$) and dry-eye symptoms, i.e., SPEED (β = 5.21, $p < 0.001$) and OSDI (β = 3.65, $p < 0.001$) scores, with an explanatory power of 57% (F = 20.72, $p < 0.001$), and those influencing ES off duty were computer usage time ≥ 4 h (β = −1.98, $p = 0.049$) and dry-eye symptoms, i.e., SPEED (β = 4.75, $p < 0.001$) and OSDI (β = 2.33, $p = 0.021$) scores, with an explanatory power of 40% (F = 10.30, $p < 0.001$; Table 3).

**Table 3.** Factors influencing perceived eye-related symptoms (n = 154).

| Variables | ES-On Duty | | | ES-Off Duty | | |
|---|---|---|---|---|---|---|
| | β | t (p) | 95 CL | β | t (p) | 95 CL |
| Intercept | | 1.94 (0.054) | −0.19 to 21.16 | | 1.76 (0.081) | −1.42 to 24.54 |
| Gender (female) | 0.18 | 3.11 (0.002) | 1.72 to 7.73 | 0.13 | 1.82 (0.071) | −0.29 to 7.02 |
| Department (COVID-19 ward) | 0.05 | 0.83 (0.406) | −1.26 to 3.11 | 0.05 | 0.69 (0.492) | −1.74 to 3.59 |
| Working environment temperature & humidity (inappropriate) | −0.04 | −0.62 (0.537) | −2.89 to 1.51 | 0.07 | 1.00 (0.319) | −1.33 to 4.06 |
| Computer usage time (≥4 h) | 0.03 | 0.49 (0.626) | −1.80 to 2.98 | −0.14 | −1.98 (0.049) | −5.85 to −0.01 |
| Diagnosis of dry eye syndrome (yes) | 0.06 | 1.05 (0.298) | −1.05 to 3.41 | −0.04 | −0.51 (0.610) | −3.41 to 2.01 |
| SPEED | 0.41 | 5.21 (<0.001) | 2.05 to 0.40 | 0.45 | 4.75 (<0.001) | 0.42 to 1.02 |
| OSDI | 0.28 | 3.65 (<0.001) | 1.88 to 0.06 | 0.21 | 2.33 (0.021) | 0.02 to 0.19 |
| Perceived health status | −0.02 | −0.39 (0.694) | 1.31 to −0.59 | 0.04 | 0.56 (0.574) | −0.43 to 0.78 |
| Occupational stress | 0.07 | 1.12 (0.265) | 1.27 to −0.06 | 0.04 | 0.59 (0.555) | −0.11 to 0.21 |
| $R^2$ (Adjusted $R^2$) | 0.57 (0.54) | | | 0.40 (0.36) | | |
| F (p) | 20.72 (<0.001) | | | 10.30 (<0.001) | | |

ES-on duty = eye-related symptoms during on duty. ES-off duty = eye-related symptoms during off duty. SPEED = Standard Patient Evaluation of Eye Dryness. OSDI = Ocular Surface Disease Index.

## 4. Discussion

This study aimed to identify the factors influencing ES in nurses during on and off-duty hours and determine the nature of these influences to provide basic data for the development of policies for management of eye health. As a predictor of ES on and off duty, dry-eye symptoms were measured using SPEED and OSDI in this study. Among the participants in this study, 72.1% had SPEED scores ≥ 6 (dry eye), which exceeded the previously reported value (45%) for day-shift healthcare workers (including physicians and nurses) [19]. Meanwhile, 23.4% of the participants in this study had OSDI scores ≥ 13 (dry eye), which was lower than the previously reported value (72%) for VDT-related workers [10]. This difference may be attributable to the fact that the work of hospital nurses involves dynamic tasks (nursing activities) rather than static tasks performed in front of the computer. The diagnosis of dry-eye syndrome was mostly based on symptoms, although questionnaire assessments are known to indicate a higher prevalence than when combined with an objective test [15]. Nevertheless, considering the mean age of the participants in this study (approximately 33 years), the incidence of dry-eye symptoms was 23.4% (OSDI ≥ 13) or 72.1% (SPEED score ≥ 6), which implies that dry eye in nurses is a health issue that demands a high level of attention.

Early detection and treatment are an effective approach to alleviate the symptoms of dry-eye syndrome and prevent further deterioration [3]. Among the participants in this study, 26% were clinically diagnosed with dry-eye syndrome, of which only 14% were receiving regular eye exams. This finding is an indirect indicator of the general low level of attention for the diagnosis and management of ES in most nurses. ES from eye dryness to reduced vision can negatively influence daily activities, e.g., by reducing reading ability or response time in driving and by reducing work productivity related to computer usage and documentation [5]. Notably, ES in on-duty nurses can interfere with tasks such as checking of prescriptions and patient details and thereby cause problems related to patient safety, indicating the need for active management. SPEED and OSDI are useful for assessing dry-eye syndrome, and based on its high accuracy, SPEED, in particular, is likely to contribute to the early assessment and efficient management of eye health issues in hospital nurses [15,16].

Female sex was identified as a factor influencing on-duty ES in this study. A study on paramedical workers in South Korea [10] and another study on VDT-related workers [20] also reported that females showed ES to a greater extent than males. Sex hormones such as androgens have been shown to have potential anti-inflammatory effects since they promote the functions of meibomian glands and secretions from lacrimal glands, and

the high sensitivity of the cornea and conjunctiva to the fluctuations of sex hormones during the menstrual cycle can influence ES [21]. The results of this study showed that sex influenced only on-duty ES, presumably because of the wearing of makeup and contact lenses during on-duty hours in addition to the participants' physiological vulnerability. Makeup on the face worn during on-duty hours can cause ES by damaging the ocular surface and tear film [22]. Additionally, the rate of contact lens usage in females is higher than that in males [23]. The mechanical or hypoxic stress induced by lenses and microbial infection of lenses can lead to eye injuries [5]. In this study, nurses showed a higher level of perceived ES during on-duty hours than during off-duty hours, presumably because of the abovementioned negative effects of on-duty use of makeup and contact lenses. According to the data obtained from Statistics Korea, 66.3% of patients with dry-eye syndrome are females [4]; likewise, the vulnerability of females to issues related to eye health has been reported to be higher than that of males [10,20]. Since most nurses are females, these findings highlight the need for greater interest in and education for eye health in women based on the greater vulnerability regarding eye health in women. Nevertheless, the percentage of male nurses in this study was approximately 12%, a relatively small sample size, and additional studies are required with a greater number of male nurses.

Computer usage time $\geq 4$ h a day was identified as a factor influencing off-duty ES in this study. This finding agrees with the results showing exacerbation of dry-eye symptoms with increased computer usage time in a previous study on healthcare professionals [10]. And a study conducted on VDT-related workers in Japan, where VDT usage $\geq 4$ h a day was strongly correlated with severe ES [24]. Computer usage leads to eye dryness by decreasing the blinking rate and accelerating tear evaporation [2]. Additionally, the prolonged use of digital devices results in digital eye strain, which is characterized by symptoms such as dry eyes, itching, a sensation of a foreign body in the eye, excessive tearing, blurred vision, and headaches [25]. It also causes symptoms such as irritation or pain in the eyes and eye tiredness [20], and exposure to the blue light of computers can damage the retina and induce eye tiredness [26]. A previous study reported that even 1–2 h of usage could induce ES [27], while another study on Spanish nurses reported that the incidence of ES was approximately 7-fold higher in nurses with VDT usage $\geq 4$ h a day [1]. In this study, 75.3% of participants reported $\geq 4$ h a day of computer usage and 46.1% reported $\geq 4$ h a day of smartphone usage. In a study investigating the work hours spent in indirect nursing tasks, including the time spent using a computer to check prescriptions and create nursing records, the mean time of computer usage was approximately 2 h [28]. Thus, the participants in this study may have been using computers for 1–2 h or more after work. Furthermore, since hospital nurses in South Korea cannot easily use smartphones during work, their smartphone usage time could be attributed to the time after work. Thus, the continuous computer and smartphone usage from on-duty to off-duty hours may have influenced the nurses off-duty ES. Since the work of nurses characteristically involves a high level of computer usage related to checking patient prescriptions and nursing records, reducing on-duty computer usage is difficult. Thus, reducing computer and smartphone usage after work can have a positive impact on ES.

To promote eye health among nurses, the working environment should be improved by adjusting the locations of computer monitors, controlling the light intensity as well as temperature and humidity to ensure the protection of nurses' eyes during computer work [3]. Additionally, for the management of digital eye strain, it is recommended to address refractive errors, including astigmatism and presbyopia, through appropriate corrective measures [29]. Appropriate education is also necessary to increase awareness of additional computer usage after work as a factor that could affect eye health, allowing nurses to regulate and improve their usage based on the duration and conditions [3]. Furthermore, it would be beneficial for nurses' eye health to engage in exercises such as reminding them to blink and stretch their bodies at regular intervals [3].

*Limitations*

This study had several limitations. First, the generalizability of the findings may be limited since the participants were recruited from a single hospital in South Korea, while the cause-effect relationship remains unknown since the study was a cross-sectional survey. Second, the symptoms on and off duty could not be clearly differentiated since the study used a retrospective analysis through recall. Third, the duration of ES and severity of symptoms and various factors of daily activities that may influence ES were not taken into account. A subjective questionnaire was used to evaluate ES in this study, and future studies should use objective eye examinations to ensure an objective estimation of ES. Moreover, in assessments of the use of VDT, VDT types other than smartphones and computers were not distinguished. Fourth, despite the variations in ES in relation to the location of work, the findings did not adequately reflect a greater variety of working environments of nurses according to their specialty (COVID-19 ward, Insurance Review Team, etc.). Fifth, no data was gathered to specifically address the issue of close-up screen usage durations when on and off duty, or the ratio of such usage while on and off duty.

## 5. Conclusions

This study identified the factors influencing the ES on and off duty in general hospital nurses. The results indicated that the level of ES experienced by general hospital nurses was higher on duty than off duty. The factors influencing ES on duty were female sex and dry-eye symptoms (SPEED and OSDI scores), whereas the factors influencing ES off duty were computer usage time and dry-eye symptoms (SPEED and OSDI scores). This study confirmed that a number of nurses experienced ES due to changes in the medical environment. The results of this study can improve awareness of eye healthcare among hospital nurses and identify the influencing factors. It can also be expanded to develop a management strategy for the health of other medical workers in the hospital. Based on the findings of this study, an eye-health management program should be developed for nurses, and the program effects should be evaluated in the future.

**Author Contributions:** O.-H.C.: Conceptualization, Methodology, Investigation, Writing—Original Draft, Writing—Review & Editing, Supervision, Project administration. H.C.: Conceptualization, Methodology, Data Collection, Writing—Original Draft. H.K.: Conceptualization, Methodology, Data Analysis, Writing—Original Draft, Writing—Review & Editing. All authors have read and agreed to the published version of the manuscript.

**Funding:** This research received no external funding.

**Institutional Review Board Statement:** The study was conducted in accordance with the Declaration of Helsinki and approved by the Institutional Review Board of the Kongju National University (No. KNU_IRB_2021-84).

**Informed Consent Statement:** Written informed consent was obtained from all participants involved in the study.

**Data Availability Statement:** Data are available upon request.

**Acknowledgments:** We give thanks to the nurses who participated in the study.

**Conflicts of Interest:** The authors declare no conflict of interest.

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
