# Peer review of "Perceived Eye-Related Symptoms and Influencing Factors in Hospital Nurses"

_healthcare, doi:10.3390/healthcare11101519_

Round 1
Reviewer 1 Report
This present study identified factors influencing eye-related symptoms in hospital nurses based on standardized questionnaires. The authors found greater levels of symptoms when the nurses are on duty, which is very relevant in informing the hospital management to look into the workplace eye and visual health of their nurses. However, the harmful effect of blue light from VDU remains highly controversial (Vera, et al, Clin Exp Optom, 2022) especially when the authors are recommending blue light protective glasses in a large scale to the nurses to combat digital eyestrain. Conversely, a more appropriate and widely acceptable recommendation would be refractive correction (i.e. prescription spectacles or contact lenses) because the nurses do spend a lot of time viewing devices, reading prescriptions, etc, as noted in the manuscript. In fact, uncorrected refractive error is a major contributor to eyestrain particularly digital eyestrain (Coles-Brennan, et al, Clin Exp Optom, 2019) rather than dry eyes alone (e.g. Al-Mohtaseb, et al, Clinical Ophthalmology, 2021). This is especially so for presbyopia, which is causes age-related blurred vision at near (e.g. Katz, et al, Clinical Ophthalmology, 2021), and astigmatism which is frequently uncorrected in the general population due to neural adaptation (e.g. Yap et al, Investigative Ophthalmology and Visual Science, 2019). Hence, this manuscript will need to be amended before it can be published. In addition, Lines 204-212 is repetition of the methods, and should be summarized in a few words instead.
Reviewer 2 Report
General comment:
Most of the sentences are too long. Suggestion to rephrase the sentences using active voice. The repeated use of the wording, COVID-19 pandemic presents it as the main focus for this paper, whereas it is not. Have a think around the main focus of the study and write strictly in that line. In some instances, the paragraphs are not structured in a way to transition to related topic sentences. There are so much unnecessary abbreviations, e.g., ES, OR, PES, ES off duty, etc. Could the authors just spell out these words?
Lines 35 – 38:
COVID-19 social distancing can no longer be considered recent. Rephrase this sentence to bring out the main message that ‘measures to limit the spread COVID-19 has generally led to a lifestyle involving long hours of indoor 36 activities, increasing the use of mobile devices and image displays even during off-duty 37 hours’
Lines 38 – 41:
Move the sentence starting with “Continuous mask wearing…..” to come after the next sentence on Lines 41 – 45. And rephrase the sentence in active voice, delete COVID-19 pandemic.
Lines 41 – 49:
Rephrase these sentences to shorten the length. Cut unnecessary words, write in active voice.
Lines 49 – 51: How young are the nurses, in terms of age? Can the authors double check to make sure they’re citing the right reference here?
Line 82: what was the effect size for sample size calculation based on? Change in symptom??
Ln 91 – 92:
What questionnaires did the participants complete? Provide specific details about the questionnaires. If they’re the ones described under the heading “Instruments”, then say this and provide a linking sentence to point the questionnaires as data collection instruments.
Line 94:
What was the gift appreciation, needs to be clear.
Line 145:
Delete “real” before number.
Line 156:
What does this sentence mean? “There were 136 (88.3%) staff nurses.”
Line 167 and Table 2:
Rephrase subheading and Table 2 title to make more sense. Table should be able to stand alone outside of the main manuscript text.
Lines 209 – 212:
Delete this sentence as it doesn’t seem to be true.
Line 259:
Delete “Lastly”, start the sentence with “Computer usage….”
Line 260”
Replace “coincided” with ‘agrees’ or ‘compares’
Line 266:
Provide a valid reference to show that blue light emission from the computers damages the retina and induce eye tiredness. The reference #10 provided does not seem to be the right reference here.
Line 288:
Provide examples of eye exercises that can be performed.
Lines 289 – 292:
This is somewhat a repetition of the conclusion. Delete from here and incorporate any word if need to the Conclusion section.
Most sentences are too long. Rephrase sentences to shorten the length. Cut unnecessary words, write in active voice
Reviewer 3 Report
It is an excelent and interesting article and may be interesting for a wide audience.
There are only some minor comments to improve the article:
It might be mentioned as limitation that no data were obtained to precisely address the issue of the durations of close-up screen useage on and off duty, or its ratio on and off duty.
The correct spectacle correction of refractive error is more critical in people who use computer screens, tablets, or smartphones for extended periods of time during the day (such as more than 4 hours) than in people who read conventionally. A viable preventive step would be to begin eye examinations for nurses who work on a close-up screen for more than 4 hours per day, including refraction and accurate spectacle prescription, as well as assessment and treatment of dry eye disease. I would suggest to address these aspects in the discussion.
